# Association between depressive symptoms and objectively measured daily step count in individuals at high risk of cardiovascular disease in South London, UK: a cross-sectional study

Vera M Ludwig,[1] Adam Bayley,[2] Derek G Cook,[3] Daniel Stahl,[4] Janet L Treasure,[5] Mark Asthworth,[6] Anne Greenough,[7,8,9] Kirsty Winkley,[2] Stefan R Bornstein,[1] Khalida Ismail[2]

For numbered affiliations see end of article.

**Correspondence to**
Prof. Khalida Ismail;
khalida.2.ismail@kcl.ac.uk

## ABSTRACT

**Objectives** Depressive symptoms are common but rarely considered a risk factor for unhealthy lifestyles associated with cardiovascular disease (CVD). This study investigates whether depressive symptoms are associated with reduced physical activity (PA) in individuals at high risk of developing CVD.

**Design** Secondary analysis of the cross-sectional baseline data from a randomised controlled trial of an intensive lifestyle intervention.

**Setting** 135 primary care practices in South London, UK.

**Participants** 1742 adults, 49–74 years, 86% male at high (≥20%) risk of developing CVD in the next 10 years as defined via QRISK2 score.

**Outcome measures** The main explanatory variable was depressive symptoms measured via the Patient Health Questionnaire-9 (PHQ-9). The main outcome was daily step count measured with an accelerometer (ActiGraph GT3X) stratified by weekdays and weekend days.

**Results** The median daily step count of the total sample was 6151 (IQR 3510) with significant differences (P<0.001) in mean daily step count between participants with low (PHQ-9 score: 0–4), mild (PHQ-9 score: 5–9) and moderate to severe depressive symptoms (PHQ-9 score: ≥10). Controlling for age, gender, ethnicity, education level, body mass index (BMI), smoking, consumption of alcohol, day of the week and season, individuals with mild depressive symptoms and those with moderate to severe depressive symptoms walked 13.3% (95% CI 18.8% to 7.9%) and 15.6% (95% CI 23.7% to 6.5%) less than non-depressed individuals, respectively. Furthermore, male gender, white ethnicity, higher education level, lower BMI, non-smoking, moderate alcohol intake, weekdays and summer season were independently associated with higher step count.

**Conclusions** People at high risk of CVD with depressive symptoms have lower levels of PA.

**Trial registration** ISRCTN84864870; Pre-results.

## Strengths and limitations of this study

► This study is the first to look at the relationship between depressive symptoms and objectively measured physical activity in a large sample (n=1720) of individuals at high risk of developing cardiovascular disease.

► The analysis controls for potential confounding of: age, gender, ethnicity, education level, body mass index, smoking, consumption of alcohol, day of the week and season.

► The inclusion of participants with as little as one valid wear day tries to minimise bias by not excluding more depressed participants that did not wear the monitor for the requested 5 days.

► No causalities can be inferred from the cross-sectional data.

► As it is a sample from the general population, the proportion of severely depressed subjects is small, and the power to compare the severity categories of depressive symptoms on step count is limited.

and contributes considerably to years lost to disability.[2] The main lifestyle risk factors for CVD are tobacco smoking, physical inactivity, unhealthy diets and the harmful use of alcohol.[1] Estimations suggest that increasing physical activity (PA) to the WHO recommended amount would reduce the global CVD-related burden by 6%.[3] It is increasingly recognised that depression (defined here as an umbrella term for depressive disorder and depressive symptoms) is an independent risk factor for CVD[4] and is associated with increased risk of mortality after a CVD event,[5] but mental health is not routinely considered as a factor that may affect uptake of PA to improve CVD outcomes in clinical practice or intervention studies.

## INTRODUCTION

Cardiovascular disease (CVD) remains the most common cause of death worldwide: it accounts for a third of global mortality[1]

For most individuals and especially the elderly, the most accessible and safest form of PA is walking. Increased number of daily steps, independent of walking speed or intensity, is associated with improved biomedical outcomes such as lower body mass index (BMI),[6] lower cholesterol ratio and triglyceride levels[7] and a lower rate of CVD incidents overall.[8] Self-report questionnaires to assess walking are prone to over-reporting and under-reporting biases.[9] Using objective tools to assess walking, such as accelerometers or pedometers, can overcome this limitation.[10] Potential confounders of any association between depressive symptoms and walking include younger age, male gender, lower BMI and white ethnicity, which are associated with higher step count.[11–14] Education is also a potential confounder, although there have been mixed findings.[11 12 15] There are also well known time effects, namely that people walk more on weekdays than on weekends and during summer months than in winter.[14]

The relationship between depressive symptoms and step count has only been assessed in specific populations with small sample sizes, such as low-socioeconomic status Latino immigrants,[16] elderly Japanese people[17] or patients with chronic conditions such as heart failure[18 19] or chronic obstructive pulmonary disease.[20 21] Studies yield contradictory results, with some observing no association between depressive symptoms and daily step count,[19 21] while others report a negative correlation.[16–18 20] In one cross-sectional sample of healthy older adults, an inverse association between depressive symptoms (using the Goldberg Depression Scale-15) and accelerometer measured daily step count disappeared after controlling for general health and disability.[22] While a systematic review suggests reduced levels of objectively measured PA in patients with depression,[23] it is not known whether this association is present in those at high risk of CVD and taken into account important confounding such as gender and age.

We aim to describe the distribution of objectively measured average daily step count in a sample at high risk of CVD and to test the hypothesis that individuals with greater depressive symptoms are likely to have reduced daily step counts, controlling for potential confounding.

## MATERIALS AND METHODS
### Design
This is a cross-sectional analysis of the baseline data of individuals at high risk of CVD, who were participating in a randomised controlled trial (RCT) testing the effectiveness of an intensive lifestyle intervention 'MOtiVational intErviewing InTervention' (MOVE IT) to increase PA and reduce weight. The data were collected before participants were randomised.

### Setting and sampling frame
The MOVE IT study was conducted in the 12 South London Clinical Commissioning Groups (CCGs), which are the local administrative authorities for commissioning

healthcare for its residents, representing a diverse socioeconomic and ethnic population of approximately 3 million UK residents. All primary care practices with a list size of >5000 registered patients resident within these CCGs were invited to participate. Within these practices, patients with a ≥20% risk of having a fatal or non-fatal CVD event in the next 10 years as defined by the QRISK2 score (2013 version),[24] were identified from the practice register. The QRISK2 score is a validated predictive tool for identifying individuals at risk of developing a fatal or non-fatal cardiovascular event in the next 10 years based on age, smoking status, ethnicity, systolic blood pressure, ratio of total serum cholesterol to high-density lipoprotein cholesterol, BMI, family history of coronary heart disease in first-degree relative, Townsend deprivation score, treated hypertension and diagnosis of rheumatoid arthritis.[24] Further patient inclusion criteria were: age ≥40 and ≤74 years, permanent residency and fluent in English. The exclusion criteria were: known CVD; on the general practice electronic register for diabetes, chronic kidney disease, atrial fibrillation or stroke; and chronic obstructive pulmonary disease or morbid obesity (BMI >50 kg/m$^2$). Participants were recruited from 7 June 2013 to 19 February 2015. All participants gave written informed consent. Further details of the trial protocol are available.[25]

### Measures
The main explanatory variable was self-report depressive symptoms using the Patient Health Questionnaire-9 (PHQ-9).[26] The PHQ-9 scores the occurrence of each of the nine *Diagnostic and Statistical Manual of Mental Disorders*, 4th Edition criteria for major depressive disorder over the past 2 weeks on a scale from 0 (not at all) to 3 (nearly every day). The total sum score, ranging from 0 to 27, can be categorised as not depressed (0–4), mild (5–9), moderate (10–14), moderately severe (15–19) and severe depression (20–27).[26] A score of 10 or above is defined as clinically relevant depression and a recent meta-analysis reported that a cut-off score ≥10 generated a sensitivity of 77% and specificity of 85% in diagnosis for major depression.[27] In our study, the number of individuals with scores in the three top categories (10–14, 15–19 and 20–27) was low (2.9, 0.9 and 0.2%, respectively); therefore, we merged them to create one category of moderate to severe 'clinical' depression (score ≥10).

The main outcome was the average daily number of steps measured with the ActiGraph GT3X accelerometer (ActiGraph, Florida, USA), a triaxial movement sensor that has been validated on number of steps taken and is reported to record 98.5% of steps.[28] Number of steps correlated (r=0.70) with average moderate-to-vigorous physical activity (MVPA). We chose steps per day as the main outcome measure in this analysis as it was specified as the main outcome measure in the MOVE IT trial, and walking is the main form of PA in this elderly population. We also note that steps per day was highly correlated with minutes of MVPA (r=0.75). Steps also has the merit that it

is an easily understood measure that facilitates clinical application of the results. The accelerometers were given to the participants by a researcher who advised them to attach the monitor to the waist and wear it for 7 days from waking in the morning until going to bed at night, only removing it for bathing and swimming. If monitors were worn for less than 9 hours (540 min) per day, these wear days were considered missing as it suggests the participant forgot to wear the monitor for a part of the day. The aim was to generate a minimum of five valid wear days per participant to ensure the collected data represent habitual activity levels.[29] A preliminary analysis revealed that people who wore the monitor for less than 5 days exhibited significantly more depressive symptoms than those who wore it for at least 5 days, so to reduce bias, we included individuals who had less than 5 days of wear time. We therefore included all participants with at least one valid wear day in the analysis and performed a sensitivity analysis to determine whether including individuals with one wear day only altered the results.

To account for potential confounding, reported covariates of depressive symptoms and objectively measured PA (age, gender, BMI, ethnicity, day of the week, level of education,[11] season[14] smoking and alcohol intake[30]) were controlled for in the analysis. During the one-on-one interview with the researcher, participants reported age, gender and ethnicity and were weighed and measured with standardised scales.[25] Self-reported levels of educational attainment were categorised into: level 1: no formal qualifications; level 2: qualifications at the end of compulsory school education in the UK typically at age 16 years; or level 3: qualifications awarded at the end of further education typically at age 18 years. Reported smoking status was categorised as: never smoked, ex-smoker and current smoker. The Alcohol Use Disorders Identification Test (AUDIT), a 10-item structured questionnaire that categorises alcohol consumption, grouped participants into abstainers (AUDIT score=0), low risk (score 1–7) and possibly hazardous or harmful drinkers (score ≥8).[31]

## Statistical analysis

Data on daily step count were downloaded from the activity monitors using Actilife V.6.11.7.[32] The number of valid wear days (≥540 min) per participant was counted. Extreme outliers in daily step count were individually investigated to ensure plausibility and rule out device failure. This was not the case and no cases were removed from the analysis. Measurements of step count were divided into weekdays (Monday–Friday) and weekend days (Saturday and Sunday). Within these categories, the arithmetic mean of the step count for each participant was calculated, thereby creating at least one and at most two measurements per participant. Baseline characteristics were summarised as means (±SD) or percentages. As the step count data were not normally distributed, median and its IQR are presented, and non-parametric Spearman rank correlations were applied. To identify relevant confounders of step count, Mann-Whitney U and Kruskall Wallis tests were conducted to compare differences between categories of potential confounders. No adjustments for multiple testing were made in order avoid missing potentially important confounders due to an inflation of the type II error for our final model.[33]

For the main analysis, a linear mixed model (LMM) was used. LMMs can account for the dependency of repeated observations of the same subject (weekdays and weekends) and allow including all individuals with at least one outcome observation.[34] To improve the fit of the model, the step count data were log-transformed before they were entered into the model. In order to investigate the sizes of the adjusted and unadjusted effects of each predictor on step count, we introduced two models. The first model aims to describe the individual associations between each predictor and mean daily step count. We controlled for age, gender, season and day of accelerometer wear (basic confounders) and then individually entered each additional covariate: (a) depressive symptoms, (b) ethnicity, (c) education level, (d) BMI, (e) smoking status and (f) drinking of alcohol into the LMM as independent predictor for log step count, obtaining the estimated change caused by each of the predictors. In the second model, all variables were entered simultaneously to determine the mutually adjusted effects of all predictors. Age and BMI were entered as continuous fixed effects variables (centred around the mean), and the others were entered as categorical fixed effects variables. We used an unstructured covariance structure for the LMM. Assumptions of the LMM were assessed by visual inspection of the residual plots.[35] Controlling for nesting at practice level showed that the estimated variance was virtually 0, thus practice level was excluded from the final LMM. The constant for each LMM is reported alongside the coefficients of relative change for each variable. These relative differences between the log transformed step counts were back transformed to the percentage difference in step count by applying the formula $100(e^{Coefficient}-1)$. To account for multiple testing in the final LMM, the alpha-level is lowered to $\alpha=0.01$, and predictors with p values <0.05 but >0.01 are treated as trend and interpreted with caution.[36]

A sensitivity analysis was conducted to determine if wearing the accelerometer for less than 5 days alters the results by rerunning the main analyses on participants with at least five wear days (online supplementary table 1). Reanalysis of QRISK2 scores using cleaned data and the QRISK2 2013 batch calculator revealed that 155 (9%) of the included participants had a QRISK2 score <20%. We conducted a second sensitivity analysis by rerunning the main analyses including only patients with a QRISK2 score of ≥20 (online supplementary table 2) to check for potential bias. Independent sample t-tests were used to compare depression scores between subsamples (≥5 vs <5 wear days and ≥20% and <20% QRISK2). Data were analysed using IBM SPSS Statistics for Macintosh, V.24.0 was used for all analyses.

 

## RESULTS
### Population characteristics
The MOVE IT sample consisted of 135 primary care practices in the 12 South London CCGs, from which 1742 participants were eligible on screening and consented to participate. Only n=22 (1.3%) provided less than one valid wear day with ≥540 min and had to be excluded right away. Additionally, 117 participants (6.7%) provided at least one but less than the requested five full wear days. When comparing these 117 participants with those who wore the monitor for 5 days or more, we found that the former were significantly more depressed than the latter (P<0.001). If we excluded this group, this may lead to an underestimation of the effect of depression on step count; therefore, we decided to include this group, giving a final sample of 1720. The average QRISK2 score of the sample

was 25% (SD 4.9). On recalculation of QRISK2 scores, 155 patients with a score <20% had been recruited into the trial. As there was no difference in depressive symptoms compared with those with QRISK2 score ≥20% (P=0.394), we retained this subgroup in the analysis. The mean age was 69 (SD 4.1) years, the vast majority were male and of white ethnicity. Demographic and behavioural characteristics of the final sample stratified by accelerometer wear day are reported in table 1.

### Step count data
A total of 11 140 wear days from n=1720 participants were included in the analysis. These participants provided a mean of 6.5 (SD 1.2) wear days. Daily step count ranged from 105 to 21 313 steps with a median of 6151 (IQR 3510). Figure 1 shows an inverse relationship (Spearman's rho

**Table 1** Demographic characteristics and distribution of median daily step count in a randomised controlled trial assessing the effectiveness of an intensive lifestyle intervention

| Variables | | Weekdays N (%) | Median steps (IQR)* | P values† | Weekends N (%) | Median steps (IQR)* | P values† |
|---|---|---|---|---|---|---|---|
| Total study population | | 1719 (100) | 6447 (3875) | | 1645 (100) | 5676 (3737) | |
| Age | >70 | 58 (3.4) | 6434 (4175) | P=0.859 | 52 (3.2) | 5639 (4209) | P=0.454 |
| | 50–59‡ | 701 (40.8) | 6319 (4017) | | 661 (40.2) | 5784 (3985) | |
| | 60–69 | 960 (55.8) | 6485 (3738) | | 932 (56.7) | 5640 (3659) | |
| Gender | Male | 1470 (85.5) | 6665 (3910) | P<0.001 | 1402 (85.2) | 5903 (3870) | P<0.001 |
| | Female | 249 (14.5) | 5441 (3343) | | 243 (14.8) | 4590 (3143) | |
| Ethnicity | White | 1538 (89.5) | 6536 (3895) | P<0.001 | 1472 (89.5) | 5800 (3757) | P<0.001 |
| | Black/Asian/other | 181 (10.5) | 5619 (3493) | | 173 (10.5) | 4676 (3321) | |
| Education level§ | Level1 | 427 (25.3) | 6022 (3742) | P=0.001 | 399 (24.7) | 5287 (3713) | P=0.018 |
| | Level 2 | 467 (27.6) | 6391 (4070) | | 444 (27.5) | 5517 (3907) | |
| | Level 3 | 796 (47.1) | 6734 (3871) | | 774 (47.9) | 5928 (3690) | |
| Depressive symptoms (PHQ-9) | None (0–4) | 1449 (84.3) | 6656 (3855) | P<0.001 | 1394 (84.7) | 5804 (3831) | P<0.001 |
| | Mild (5-9) | 201 (11.7) | 5563 (3330) | | 189 (11.5) | 5045 (3473) | |
| | Moderate/severe (≥10) | 69 (4.0) | 5428 (3905) | | 62 (3.8) | 5258 (3939) | |
| BMI (kg/m²) | <20 | 24 (1.4) | 6899 (4774) | P<0.001 | 22 (1.3) | 6178 (3899) | P<0.001 |
| | 20–24.9 | 371 (21.6) | 7255 (3983) | | 365 (22.2) | 6307 (4329) | |
| | 25–29.9 | 800 (46.5) | 6550 (3736) | | 773 (47.0) | 5796 (3615) | |
| | ≥30 | 524 (30.5) | 5850 (3594) | | 485 (29.5) | 5108 (3559) | |
| Smoking status | Never smoked | 482 (28.0) | 6485 (3694) | P<0.001 | 468 (28.4) | 5795 (3760) | P=0.047 |
| | Ex-smoker | 975 (56.7) | 6605 (3850) | | 929 (56.5) | 5784 (3670) | |
| | Current smoker | 262 (15.2) | 5795 (4285) | | 248 (15.1) | 5268 (4115) | |
| Alcohol intake (AUDIT) | Abstainer (0) | 176 (10.2) | 5533 (4208) | P<0.001 | 171 (10.4) | 4984 (3598) | P=0.001 |
| | Low risk (1-7) | 1274 (74.2) | 6544 (3628) | | 1216 (73.9) | 5767 (3629) | |
| | Harmful (≥8) | 269 (15.6) | 6490 (4370) | | 258 (15.7) | 5787 (4027) | |

*Median step count and IQR are rounded to the closest full number.
†P values are calculated with Mann-Whitney U tests and Kruskall-Wallis tests.
‡One person was younger than 50 years (49); this person was included in the ≥50 category.
§n=29 missing for weekdays; n=28 missing for weekends.
AUDIT, Alcohol Use Disorders Identification; BMI, body mass index; PHQ-9, Patient Health Questionnaire-9.

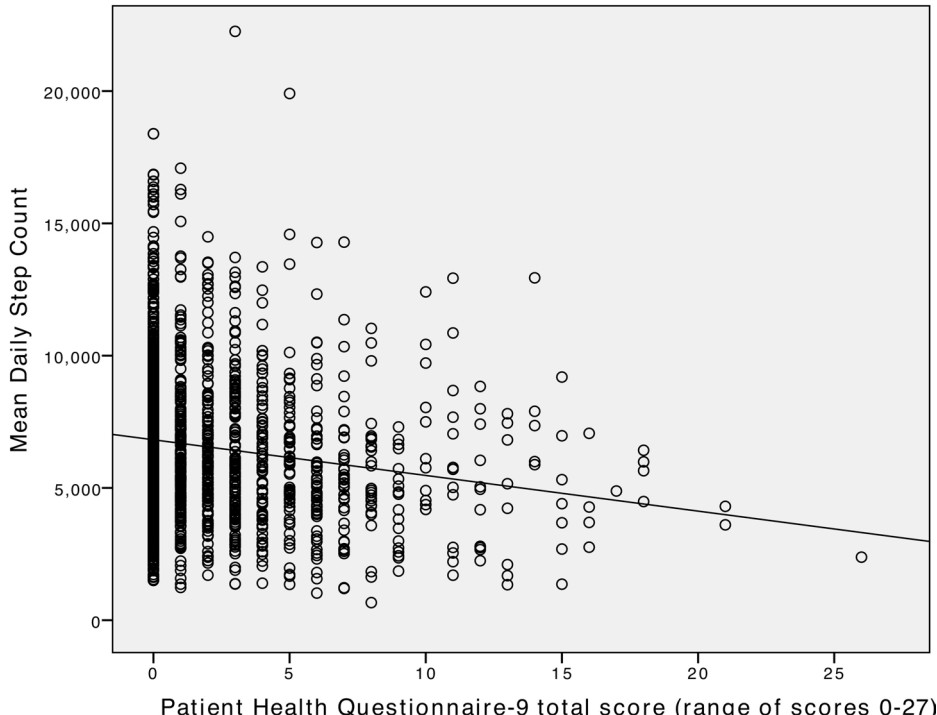

**Figure 1** The association between depressive symptoms and daily step count at baseline in adults with high cardiovascular risk in a randomised controlled trial assessing the effectiveness of an intensive lifestyle intervention to improve physical activity and reduce weight.

r=−0.149, P=0.01) between depressive symptom score and mean individual daily step count. Step count also differs between categories of gender, ethnicity, education level, BMI, smoking, drinking of alcohol and with weekdays, but not age (table 1).

### Linear mixed model

The LMMs (table 2) observed significant associations between depressive symptoms and step count. Controlling for age, gender day and season (model 1), individuals with moderate to severe depressive symptoms walked 19.0% less (P<0.001) than non-depressed individuals, and mildly depressed individuals walked 16.2% less (P<0.001). Mutually controlling for all predictors (model 2) decreased the effect of depressive symptoms to 15.6% and 13.5%, respectively (P=0.001 and P<0.001). All other covariates influenced PA as well. In model 2, each additional year in age and one point increase in BMI was associated with a decrease in step count by 1.2% and 2.7%, respectively (P<0.001 and P<0.001). Female gender reduced step count by 13.8% (P<0.001); individuals from ethnic minorities walked 17.8% less steps (P<0.001). Higher educational qualifications tended to be associated with higher step count. Subjects with level 3 qualifications walked 5.3% more than level 2 (P=0.028) and 5.7% more than those without formal qualification (P=0.022). Current smokers' step count was 19.1% lower compared with people who had never smoked (P<0.001), and participants abstaining from alcohol consumption had an 8.8% lower step count than people who are low risk drinkers (P=0.01). On weekends, 14.8% fewer steps were recorded

than on weekdays (P<0.001), and during the winter months, there was a decrease of 9.3% compared with spring (P=0.001). Sensitivity analyses revealed that rerunning the models with participants with at least 5 days of complete wear time (online supplementary table 1) and participants with QRISK2 score >20 only (online supplementary table 2) provided similar parameter estimates and did not alter the conclusions of this study.

## DISCUSSION
### Main study findings

This study assessed patterns of PA and depressive symptoms in people at high risk of CVD. We found an inverse relationship where increased depressive symptoms were associated with significantly decreased step counts. Female gender, ethnic minorities, lower education level, higher BMI, smoking, alcohol abstainers, weekends and colder season were independently associated with lower daily step count.

### Study strengths and limitations

The strengths of our study are, first, the large sample size. Furthermore, PA was objectively assessed based on accelerometer data. Sociodemographic, biomedical and depression data were collected in face-to-face interviews with trained researchers using standardised schedules, resulting in very few missing data. Recruitment was conducted year-round so seasonal variations were accounted for.

**Table 2** Estimated effects (coefficients and relative % change) of individual characteristics on average daily steps in subjects at high risk of CVD

| Variables | Model 1† | | Model 2‡ | |
|---|---|---|---|---|
| | Coefficient | Relative % Change (95% CI) | Coefficient | Relative % Change (95% CI) |
| Constant | 8.813 | | 8.937 | |
| **Basic confounders** | | | | |
| Age (impact additional year) | 0.004 | 0.4 (−0.1 to 1.0) | −0.012*** | −1.2 (−1.7 to −0.6) |
| Gender | | | | |
| Male | 0 | 0 | 0 | 0 |
| Female | −0.249*** | −22.1 (−26.7 to −17.2) | −0.149*** | −13.8 (−18.9 to −8.5) |
| Day | | | | |
| Weekday | 0 | 0 | 0 | 0 |
| Weekend | −0.161*** | −14.9 (−16.6 to −13.2) | −0.161*** | −14.9 (−16.6 to −13.1) |
| Season | | | | |
| Spring | 0 | 0 | 0 | 0 |
| Summer | −0.007 | −0.7 (−6.7 to 5.7) | 0.004 | 0.4 (−5.4 to 6.7) |
| Autumn | −0.013 | −1.3 (−7.2 to 5.0) | −0.015 | −1.5 (−7.2 to 4.5) |
| Winter | −0.086** | −8.3 (−13.8 to −2.4) | −0.098** | −9.3 (−14.5 to −3.8) |
| **Added covariates** | | | | |
| a) Depressive symptoms constant | 8.84 | | | |
| None (PHQ-9: 0–4) | 0 | 0 | 0 | 0 |
| Mild (PHQ-9: 5–9) | −0.177*** | −16.2 (−21.5 to −10.6) | −0.142*** | −13.3 (−18.6 to −7.6) |
| Moderate/Severe (PHQ-9 ≥ 10) | −0.211*** | −19.0 (−27.2 to −10.0) | −0.170** | −15.6 (−23.8 to −6.6) |
| b) Ethnicity constant | 8.845 | | | |
| White | 0 | 0 | 0 | 0 |
| Black/Asian/other | −0.145*** | −13.5 (−19.2 to −7.4) | −0.196*** | −17.8 (−23.3 to −12.0) |
| c) Education level constant | 8.845 | | | |
| Level 3: A Level or higher | 0 | 0 | 0 | 0 |
| Level 2: O Level/GCSE/NVQ | −0.063* | −6.1 (−10.7 to −1.2) | −0.054* | −5.3 (−9.7 to −0.6) |
| Level 1: no formal qualification | −0.072** | −7.0 (−11.7 to −2.0) | −0.196* | −5.7 (−10.4 to −0.9) |
| d) BMI constant | 8.811 | | | |
| Impact of one unit BMI increase | −0.025*** | −2.4 (−2.9 to −1.9) | −0.027*** | −2.7 (−3.2 to −2.2) |
| e) Smoking status constant | 8.849 | | | |
| Never smoked | 0 | 0 | 0 | 0 |
| Ex-smoker | −0.023 | −2.3 (−6.9 to −2.5) | −0.038 | −3.7 (−8.2 to 0.9) |
| Current smoker | −0.141*** | −13.2 (−18.9 to −7.0) | −0.212*** | −19.1 (−24.4 to −13.4) |
| f) AUDIT score constant | 8.824 | | | |
| Low risk (score 1 – 7) | 0 | 0 | 0 | 0 |
| Possibly harmful (score ≥ 8) | −0.001 | −0.1 (−5.8 to 6.0) | 0.032 | 3.3 (−2.4 to 9.3) |
| Abstainer (score 0) | −0.124** | −11.7 (−17.6 to −5.2) | −0.092* | −8.8 (−14.9 to −2.2) |

Significance levels: *P<0.05, **P<0.01 and ***P<0.001.
†Model 1: all variables (a-f) adjusted for basic confounders (age, gender, day and season) only. Note: there is a constant for the basic confounders only model and separate constants for this model adjusted for each added covariate (a–f).
‡Model 2: all variables are mutually adjusted; thus, there is only one constant for the whole model.
The reference group within each category is set to 0.
AUDIT, Alcohol Use Disorders Identification Test; BMI, body mass index; CVD, cardiovascular disease; GCSE, General Certificate of Secondary Education; NVQ, National Vocational Qualification; PHQ-9, Patient Health Questionnaire-9.

The limitations are that the study design is cross-sectional, so causal relationships cannot be inferred, and it remains unclear whether people with depressive symptoms are at increased risk of walking less or less active people are at increased risk for depressive symptoms. Although PA was measured objectively using accelerometers, we cannot rule out that the recorded levels of PA are subject to bias, as participants might have changed their performance because they knew they were being observed.[37] This would not, however, have biased estimates of the association with depressive symptoms unless depressed individuals respond differently to wearing accelerometers. As this is the baseline sample of an RCT to improve CVD risk by increasing PA and reducing weight, there is the risk of a participation bias in our sample. The participants may be more interested in exercise than the general population, as they have consented in taking part in a behaviour change study. The percentage of female participants was small (14.5%), reflecting the lower QRISK2 sores in females.[24] However, data on females with high CVD risk is relevant as the prevalence of depression is higher in females, and they have worse outcomes following CVD events.[38]

The number of study participants with severe depressive symptoms was small, which is to be expected in a primary care population. Our analysis showed that depressive symptom scores were significantly higher in individuals who failed to provide the requested five wear days. We tried to minimise any measurement bias by depression by including all participants with as little as one valid wear day. Finally, as in any study with missing outcome data, we cannot exclude potential bias if missing values do depend on unobserved values (not missing at random).[39]

## Comparison with other studies

Normative data suggest that between 2000 and 9000 daily steps are a typical estimate of usual daily activity levels in healthy older adults (≥50 years).[13] Our sample falls into that range with an average of 6515 daily steps. The high inter-individual variability in step count, in our case an IQR of 3510, is also similar to findings in previous studies.[11 12 15] Women seem to be especially likely to have a sedentary lifestyle, particularly on weekends. Ethnicity and BMI showed a significant influence on step count, as reported in previous studies.[11 13 14] The role of education level for PA continues to be uncertain with significant differences in daily steps between education levels in model 1 but only trend level significance in model 2. Similarly, we found that individuals with moderate alcohol consumption walked more than abstainers (significant difference in model 1, trend level significance in model 2), which has been reported previously in accelerometer and large population-based studies.[40 41] One speculation is that these participants might be what is termed 'social drinkers' who are perhaps more likely to travel for a drink, for example to the pub, whereas abstainers might have an advanced disease or are socially isolated. These findings suggest prevention programmes to increase PA should be targeted at specific subgroups of people at high risk of CVD, as well as the general population.

The prevalence of clinically relevant depressive symptoms (PHQ-9 score ≥10) in our sample was 4%, which is midway between the reported UK prevalence of 2.8% in the general population (2.6% in males, 3.0% in females)[38] and 7% in patients with CVD.[42] It is important to note that these statistics refer to current episodes of depressive symptoms (within the past 2 weeks). Lifetime prevalence of depression in the UK is reported to be higher, at around 25.8%,[43] and may also be associated with reduced PA to a greater or lesser extent, but the difference in PA levels between the two constructs of depressive symptoms have not yet been examined.

Our participants did not have CVD and were not on clinical primary care registers for other long-term conditions. However, they were at high risk of developing CVD, and the prevalence rate of depressive symptoms was higher than in the general population. This is consistent with the results of a recent Danish register study which reported that depression might be an antecedent for CVD events and mortality,[44] further indicating the importance of depressive symptoms as an easily measured risk factor for CVD. Previous studies on the association between objectively measured PA and depression yielded inconclusive results. They have reported either a lack of a relationship,[19 21] an inverse association[16–18 20] or an inverse association that disappeared when controlled for confounding.[22] The studies that reported negative associations between daily step count and depressive symptoms have been conducted in relatively small (n=96 to n=207)[17–19 22] or in highly selective samples such as severe COPD.[20] Our study is the first to examine this relationship in a large sample of community-dwelling older European adults at high risk of CVD who were not known to have a chronic disease.

## Implications for clinicians

Practitioners and researchers should consider assessing depressive symptoms when assessing patients' CVD risk or their barriers to uptake of lifestyle interventions. Even mild symptoms of depression (PHQ-9 scores 5–9) were associated with a significantly reduced average daily step count of 13.3%. Similarly, it has been shown that mild symptoms of depression increase mortality risk after acute myocardial infarction.[45] A Cochrane review and a meta-meta-analysis on the effectiveness of PA interventions for the treatment of depression in clinical[46] and non-clinical samples[47] summarised that there appeared to be a moderate positive effect of increased PA on depressive symptoms. However, when stratified by quality of studies, the Cochrane review found no statistically significant evidence in methodologically robust RCTs that exercise was more effective than psychological or pharmacological therapies.[46] Nevertheless, we still found that depressive symptoms are associated with lower PA levels in individuals at high risk of CVD, which highlights the importance of screening and optimising conventional

depression management[48] to reduce depressive symptoms, which could help lower CVD risk.[3 4]

## Implications for future work

The varying patterns of PA in individuals at high risk of CVD indicate that it might be most efficient to target PA interventions specifically towards those with lowest activity ratings, for example, females, those from African, Caribbean or Asian ethnicities, those with higher BMIs or current smokers. Additionally, our results suggest that there is a moderate association between depressive symptoms and daily step count in individuals at high risk of CVD and that this should be considered as a potential modifying factor in clinical trials for lifestyle interventions in CVD research.

**Author affiliations**
[1]Department of Medicine III, University Hospital Carl Gustav Carus, Technical University Dresden, Dresden, Germany
[2]Department of Psychological Medicine, Institute of Psychiatry, Psychology and Neuroscience, King's College London, London, UK
[3]Population Health Research Institute, St George's, University of London, London, UK
[4]Department of Biostatistics and Health Informatics, Institute of Psychiatry, Psychology and Neuroscience, King's College London, London, UK
[5]Section of Eating Disorders, Institute of Psychiatry, Psychology and Neuroscience, King's College London, London, UK
[6]Department of Primary Care and Public Health Sciences, King's College London, London, UK
[7]MRC & Asthma UK Centre in Allergic Mechanisms of Asthma, King's College London, London, UK
[8]Department of Women and Children's Health, School of Life Course Sciences, Faculty of Life Sciences and Medicine, King's College London, London, UK
[9]NIHR Biomedical Research Centre at Guy's and St Thomas' NHS Foundation Trust and King's College London, London, UK

**Acknowledgements** We would like to thank the research team, healthy lifestyle facilitators, patient volunteers and the South London Clinical Research Network, participating general practices and their patients. We would also like to thank the Trial Steering Committee: Professor Steve Iliffe (chair), University College London; Professor Tom Marshall, University of Birmingham; Professor James Carpenter, London School of Hygiene & Tropical Medicine, MRC Clinical Trials Unit; and Dr Tim Anstiss, independent medical doctor and consultant. We would also like to thank the Data Ethics and Monitoring Committee: Professor Betty Kirkwood (previous chair), London School of Hygiene & Tropical Medicine and Professor Helen Weiss (chair), London School of Hygiene & Tropical Medicine; Professor Stephanie Taylor, Queen Mary University of London, Barts and The London School of Medicine and Dentistry, Blizard Institute; and Dr David Blane, Imperial College London.

**Contributors** KI drafted the initial hypothesis. VML developed the protocol, especially supported by AB who, as trial manager, had knowledge of all study particulars and had helped acquire the data. Furthermore, VML wrote the first draft of the report and performed the literature search. Data analysis was conducted by DS, DGC and VML. JLT, MA, AG and KW helped interpret the data and gave important feedback in the revision process. SRB contributed to the research funding, and KI and SRB coordinated the research and oversaw the project. All authors critically reviewed the manuscript for important intellectual content and approved the final manuscript.

**Funding** The MOVE IT study is funded by the National Institute for Health Research - Health Technology Assessment Programme (NIHR-HTA) (Project: 10/62/03), and work will be published in the *Health Technology Assessment* journal. The research attachment of VML to King's College London in London, UK, was funded by the International Network on Diabetes and Depression (IN2D), which is financed by the German Federal Ministry of Education and Research (BMBF). DS and KI received partial funding by the National Institute for Health Research (NIHR) Biomedical Research Centre at South London, and Maudsley NHS Foundation Trust and King's College London.

**Disclaimer** The views expressed are those of the author(s) and not necessarily those of the NHS, the NIHR or the Department of Health.

**Competing interests** KI discloses that she has received honorarium from Eli-Lilly, Sanofi, Janssen and Sunovion for lectures at educational events. However, she states that this does not constitute a conflict of interest for the present study.

**Patient consent** Obtained.

**Ethics approval** Ethical approval was granted by the Dulwich Ethics Committee (reference: 12/LO/0917).

**Provenance and peer review** Not commissioned; externally peer reviewed.

**Data sharing statement** The full data set and statistical code are available from the corresponding author Professor Khalida Ismail at khalida.2.ismail@kcl.ac.uk upon reasonable request.

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
