## [Reviewer comments · BMJ Open]

ARTICLE DETAILS

TITLE (PROVISIONAL)	The association between depressive symptoms and objectively measured daily step count in individuals at high risk of cardiovascular disease in south London, United Kingdom: a cross-sectional study
AUTHORS	Ludwig, Vera; Bayley, Adam; Cook, Derek; Stahl, Daniel; Treasure, Janet; Asthworth, Mark; Greenough, Anne; Winkley, Kirsty; Bornstein, Stefan; Ismail, Khalida

VERSION 1 – REVIEW

REVIEWER	Caroline Vandeleur Lausanne University Hospital, Department of Psychiatry, Prilly, Switzerland
REVIEW RETURNED	19-Dec-2017

GENERAL COMMENTS	This manuscript has an objective measure of daily step counts (actigraphy) in individuals at a high-risk of developing cardiovascular diseases (CVD) and the analysis assessing the association between depressive symptoms and step counts adjusts for a wide array of covariates that were nearly all initially associated with the step counts. The manuscript is well-written, the type of analysis appears to be correct, the conclusions seem founded on the data as they stand and most of the limitations are clearly stated. However, I have a concern with the assessment of depressive symptoms in this manuscript. The authors used the PHQ-9 based on the DSM-IV criteria. The PHQ-9 assesses symptoms over the past 2 weeks. The authors used a cut-off score of more than 10 to qualify moderate to severe depression scores as suggested in the publication by Manea et al. 2015. (I must add that this publication only compares the cut-off score of >10 to the algorithmic scoring method of the same instrument). The authors report a prevalence estimate of 4% of the sample with a score of >10 according to the PHQ-9. They report that this estimate is “midway between the UK prevalence of 2.8% in the general population and 7% in CVD patients”. As the vast majority of epidemiological studies that used diagnostic interviews, and not questionnaire-based evaluations, reported lifetime prevalence estimates ranging from 14.6% (World Mental Health Survey Initiative – high-income countries) to 19.2% in the US (Bromet et al. 2011) and even up to 32.5% in the Zurich study (Angst et al., 2016) and 43% in the PsyCoLaus study (Vandeleur et al., 2017), their prevalence rate still seems low and needs more explanation. In the PsyCoLaus study, about 5.6% of the sample reported having a current episode of depression, at the time of the interview. If the PHQ-9 assesses symptoms over the last 2
---

	weeks, it does not seem to potentially adequately capture all the depressions. The use of questionnaires for the assessment of depression leads to an under-estimation of depression compared to when direct interviews are used, only a percentage of the depressions are seemingly captured using this method. For this reason, I think that a paragraph on this limitation needs to be added and discussed in the discussion section. The authors should indicate how the under-estimation of depressive symptoms may have impacted their results and conclusions. References: Bromet, E., Andrade, L.H., Hwang, I., Sampson, N.A., Alonso, J., de Girolamo, G., de Graaf, R., Demyttenaere, K., Hu, C., Iwata, N., Karam, A.N., Kaur, J., Kostyuchenko, S., Lepine, J.P., Levinson, D., Matschinger, H., Mora, M.E., Browne, M.O., Posada-Villa, J., Viana, M.C., Williams, D.R., Kessler, R.C., 2011. Cross-national epidemiology of DSM-IV major depressive episode. BMC Med. 9, 90. Angst, J., Paksarian, D., Cui, L., Merikangas, K.R., Hengartner, M.P., Ajdacic-Gross, V., Rossler, W., 2016. The epidemiology of common mental disorders from age 20 to 50: results from the prospective Zurich cohort Study. Epidemiol Psychiatr Sci. 25, 24-32. Vandeleur, C.L., Fassassi, S., Castelao, E., Glaus, J., Strippoli, M.-P.F., Lasserre, A.M., Rudaz, D., Gebreab, S., Pistis, G., Aubry, J.-M., Angst, J., Preisig, M., 2017. Prevalence and correlates of DSM-5 major depressive and related disorders in the community. Psychiatry Research, 250, 50-58. Other issues are as follows:  - More detail on the assessment of the confounders (covariates) should be provided in the methods section. How was this information gathered, using which type of assessments? - Could the authors explain why female gender appears to be associated with reduced step counts in the models when the raw data suggest that it is rather male gender which reveals the lower step count? Is there an error? - Could the authors give an explanation as to why individuals with a harmful alcohol consumption show a higher step count than those abstaining from alcohol? - Could the authors provide a description of the QRisk2 score and how it is used, within the present manuscript? This would enable the reader to better understand the characteristics of the sample without having to refer to previous publications. How was it known that the subjects at high risk of CVD did not have chronic diseases? What criteria then determined their "high-risk" for CVD? A more complete description is warranted. - Whereas the manuscript is well-written overall, there are a few English mistakes and typos here and there and the manuscript could still benefit from a re-read.
--	---

REVIEWER	Amanda Rebar Central Queensland University, Australia
REVIEW RETURNED	21-Dec-2017
GENERAL COMMENTS	I think the state of the literature is such that a cross-sectional test of the association of physical activity (even objectively monitored) and depressive symptoms no longer advances our knowledge. There is

	quite a bit already known about the association of physical activity and depressive symptoms (see Cooney et al. 2014 Cochrane review; and Rebar et al., 2016 meta-meta-analysis in Health Psychology Review), which includes several studies with similar populations. However, there is exciting potential for this study to provide more rich evidence of prospective, intervention or mediation evidence relevant for the link between activity and depression in people at risk for CVD. For example, previous studies have shown that the association between depressive symptoms and adverse CVD events was explained by physical inactivity (Whooley et al., 2008, JAMA). By using more of the RCT data points, this study could provide for important evidence The conclusion of the abstract includes a statement that ‘this may potentially modify the effectiveness of lifestyle interventions to reduce the risk of CVD.’ This seems a bit of a stretch based on the data presented in this study. Please consider a conclusion more closely tied to the study findings. Given that the measure of depressive symptoms is continuous, why not test it as a continuous measure instead of separated into categories based on scores? The point about people with more depressive symptoms tended to wear the monitors for less days than people without depressive symptoms is an important one and the authors should be commended for considering the potential bias this may cause. Please clarify what is meant by ‘merging’ the step counts for weekend and weekdays. For the t-tests, adjustments to the p-values are needed for the multiple test comparisons. The value of the categories for the covariate variables need to be presented. Was there nesting at the level of clinic that should perhaps be accounted for? Check the intraclass correlations for that. The discussion often talks about change (e.g., ‘decrease’ ‘worsening’), which does not reflect the cross-sectional association found.
--	---

REVIEWER	Dr Caroline Anne Mitchell ACADEMIC UNIT OF PRIMARY MEDICAL CARE, UNIVERSITY OF SHEFFIELD
REVIEW RETURNED	31-Dec-2017

GENERAL COMMENTS	Overall this a well written, novel and interesting study investigating the interplay between physical activity and depression within a group of people at higher risk of cardiovascular events. The limitations section should be strengthened with more commentary on the biases of the study.  1) spelling error in abstract: practises should be ‘practices’ plural of practice noun not verb 2) The abstract is clear and the method used to explore associations robust 3) ERROR IN TABLE OR MAIN TEXT RE GENDER BALANCE OF THE SAMPLE Population characteristics: authors to check the statement the ‘vast majority were male’ made within this paragraph as Table 1 (total population characteristics) states that 85.5% + were female and 14.5% were men. I think the table must be incorrect as in the discussion, the explanation for the gender bias of the sample is that women are less likely to have a raised q-risk. If table 1 is incorrect then the reader needs to be assured that the rest of the tables are correct in relation to the
---

	statistical analyses by gender as a confounder. 4) The limitations section I think should be expanded and more explicitly acknowledge the biases within a population derived from a sample recruited to a RCT : These limitations include: a. the generalizability of the sample . The observations are from the baseline data of participants in a RCT. There is therefore risk of participation bias as the sample participating in the study will differ to an unmeasured extent from the rest of the primary care population within the capital city of the UK. This population may well already be more likely to be interested in exercise having consented to participate in an activity monitoring behaviour change study. b. The rationale to merge all categories of depression with a PHQ9> 10 into a single category was necessary as a relatively small number of people self reported potentially clinically significant depression with PHQ9 score > 10 . Thus the total within the potentially clinically significant depression category was 4% of the total population c. Population characteristics- Table 1 states that 8.5 5%+ were women , almost 89.5% were white which does not reflect the general population of the UK, and especially not the ethnic diversity of London. So the following statement , I think should be changed to acknowledge this bias The statement: is representative of the ethnic, socioeconomic, urban and semi-rural diversity typical of large cities in the developed nations. As 2011 census data states that : London had a population of 8,173,941. Of this number, 44.9% were White British. 37% of the population were born outside the UK, including 24.5% born outside of Europe.
--	---

REVIEWER	Patricia Wong University of Pittsburgh, USA
REVIEW RETURNED	03-Jan-2018

GENERAL COMMENTS	The authors present an interesting and relevant question: does depression predict low physical activity among those at risk for cardiovascular disease? Because increasing physical activity can help reduce this risk, understanding what may hinder improved activity levels can have important implications for treatment (i.e., targeting depression if it is a risk factor). There are several comments and questions this reviewer has concerning the methods and interpretation of results: 1. A strength of the study is the large sample size and activity measured via the Actigraph. In addition, the authors take into account several important control variables. One variable not considered is sleep-- do the authors have access to information regarding participants sleep characteristics (duration, quality, timing, etc) either measured via the same Actigraph data or through other measures? Because sleep disturbances can be related to depression, physical activity, and cardiovascular disease risk, it'd be
--

	important to also consider this additional control variable. 2. The "Implications for clinicians" statement is a bit unclear. The authors conclude that depression may be an important factor to consider when treating those with cardiovascular risk because it may hinder interventions focused on improving physical activity. However, they cite evidence that shows targeting physical activity does not improve depression symptoms. This rationale, at least as written, does not support the conclusion: ineffectiveness of physical activity interventions to treat depression does not indicate that depression hinders the treatment of physical activity. As the authors noted, their data is cross-sectional so any directionality between depression, physical activity, and effectiveness of treatment for cardiovascular risk cannot be assumed. Is there other previous literature the authors can draw from to support their conclusion?
--	---

VERSION 1 – AUTHOR RESPONSE

Responses to reviewer comments

Thank you all for your helpful and constructive comments and concerns. We will address them individually following each comment:

Reviewer: 1

Reviewer Name: Caroline Vandeleur

Institution and Country: Lausanne University Hospital, Department of Psychiatry, Prilly, Switzerland

Please state any competing interests: None declared

Please leave your comments for the authors below

This manuscript has an objective measure of daily step counts (actigraphy) in individuals at a high-risk of developing cardio-vascular diseases (CVD) and the analysis assessing the association between depressive symptoms and step counts adjusts for a wide array of covariates that were nearly all initially associated with the step counts. The manuscript is well-written, the type of analysis appears to be correct, the conclusions seem founded on the data as they stand and most of the limitations are clearly stated.

Point 1:

However, I have a concern with the assessment of depressive symptoms in this manuscript. The authors used the PHQ-9 based on the DSM-IV criteria. The PHQ-9 assesses symptoms over the past 2 weeks. The authors used a cut-off score of more than 10 to qualify moderate to severe depression scores as suggested in the publication by Manea et al. 2015. (I must add that this publication only compares the cut-off score of >10 to the algorithmic scoring method of the same instrument). The authors report a prevalence estimate of 4% of the sample with a score of >10 according to the PHQ-9. They report that this estimate is "midway between the UK prevalence of 2.8% in the general population and 7% in CVD patients". As the vast majority of epidemiological studies that used diagnostic interviews, and not questionnaire-based evaluations, reported lifetime prevalence estimates ranging from 14.6% (World Mental Health Survey Initiative – high-income countries) to 19.2% in the US (Bromet et al. 2011) and even up to 32.5% in the Zurich study (Angst et al., 2016) and 43% in the PsyCoLaus study (Vandeleur et al., 2017), their prevalence rate still seems low and needs more explanation. In the PsyCoLaus study, about 5.6% of the sample reported having a current episode of depression, at the time of the interview. If the PHQ-9 assesses symptoms over the last 2 weeks, it does not seem to potentially adequately capture all the depressions. The use of questionnaires for the assessment of depression leads to an under-estimation of depression compared to when direct interviews are used, only a percentage of the depressions are seemingly

captured using this method. For this reason, I think that a paragraph on this limitation needs to be added and discussed in the discussion section. The authors should indicate how the under-estimation of depressive symptoms may have impacted their results and conclusions.

References:

Bromet, E., Andrade, L.H., Hwang, I., Sampson, N.A., Alonso, J., de Girolamo, G., de Graaf, R., Demyttenaere, K., Hu, C., Iwata, N., Karam, A.N., Kaur, J., Kostyuchenko, S., Lepine, J.P., Levinson, D., Matschinger, H., Mora, M.E., Browne, M.O., Posada-Villa, J., Viana, M.C., Williams, D.R., Kessler, R.C., 2011. Cross-national epidemiology of DSM-IV major depressive episode. *BMC Med.* 9, 90.

Angst, J., Paksarian, D., Cui, L., Merikangas, K.R., Hengartner, M.P., Ajdacic-Gross, V., Rössler, W., 2016. The epidemiology of common mental disorders from age 20 to 50: results from the prospective Zurich cohort Study. *Epidemiol Psychiatr Sci.* 25, 24-32.

Vandeleur, C.L., Fassassi, S., Castelao, E., Glaus, J., Strippoli, M.-P.F., Lasserre, A.M., Rudaz, D., Gebreab, S., Pistis, G., Aubry, J.-M., Angst, J., Preisig, M., 2017. Prevalence and correlates of DSM-5 major depressive and related disorders in the community. *Psychiatry Research*, 250, 50-58.

Reply:

Two main issues are addressed here: a) lifetime vs. current depressive symptoms and b) measuring depression with questionnaires vs. with diagnostic interviews.

a) It is important to note that we always referred to current episodes of depression in our manuscript and not lifetime prevalence of depression. Both studies that are referred to by the reviewer (UK prevalence of 2.8% in Paykel et al., 2005 and prevalence in CVD patients 7% in Walters, et al., 2014) used diagnostic interviews to assess current depressive symptoms (defined as within the past two weeks) with the Clinical Interview Schedule-Revised (CIS-R). Moreover, in the PsyCoLaus study current depressive symptoms were 5.6%. Thus, with all due respect, the results that assess current depressive symptoms are very similar to our result (4% report current clinically relevant symptoms of depression) and we do not think the PHQ-9 failed to adequately assess depression in our study. We also took the opportunity to review the literature again and we could not identify a study that tested the association between lifetime depression and PA and we have made a note of this in the discussion (lines 344-349).

b) In the Zurich cohort study, the past year prevalence of MDD in the first year was 4% as assessed by the SPIKE interview (Angst et al., 2015) and 5.5% as assessed via CIDI in ten high income countries combined (Bromet et al., 2011). Considering that these are results from representative population samples over 12 months assessed via interviews, they seem similar to our result of 4% current depressive symptoms in a sample of individuals at high risk of CVD as measured by the PHQ-9. One might expect slightly higher levels of depression in individuals at high risk of CVD than in general population, but of course lower prevalence for current episodes than for 12-month prevalence. Thus, in our opinion, employing the PHQ-9 did not seem to lower the detection rate of depressive symptoms in our sample.

We would also like to clarify to the editor that we used the PHQ-9 cut of ≥ 10 which is the validated and conventional cut off for depressive disorder caseness and not as reviewer 1 states > 10 .

Point 2:

More detail on the assessment of the confounders (covariates) should be provided in the methods section. How was this information gathered, using which type of assessments?

Reply:

We added further information about the types of assessments in the methods section (lines 177 to 186).

Point 3:

Could the authors explain why female gender appears to be associated with reduced step counts in the models when the raw data suggest that it is rather male gender which reveals the lower step count? Is there an error?

Reply:

Thank you for noticing this error. The labels 'male' and 'female' in Table 1 were swapped by mistake. The corrected findings suggest that male participants walked more steps, as the reviewer noted.

Point 4:

Could the authors give an explanation as to why individuals with a harmful alcohol consumption show a higher step count than those abstaining from alcohol?

Reply:

Previous studies have already reported this finding in accelerometer-based studies (Westerterp et al., 2004) and in large, population based samples (French et al., 2009). Our hypothesis is that individuals who drink more are more social and might go out more (e.g. to the pub). We added this to the discussion (lines 333 to 338).

Point 5:

Could the authors provide a description of the QRisk2 score and how it is used, within the present manuscript? This would enable the reader to better understand the characteristics of the sample without having to refer to previous publications. How was it known that the subjects at high risk of CVD did not have chronic diseases? What criteria then determined their "high-risk" for CVD? A more complete description is warranted.

Reply:

More detailed description of the QRisk2 score and the factors that go into the calculation are now added to the methods section (Lines 127 to 132). Exclusion criteria are described afterwards in which we explain that potential participants who were on the general practice electronic register for a number of chronic diseases were excluded (lines 133 to 139).

Point 6:

Whereas the manuscript is well-written overall, there are a few English mistakes and typos here and there and the manuscript could still benefit from a re-read.

Reply:

Thank you, we double-checked spelling and language and corrected a few small mistakes.

Reviewer: 2

Reviewer Name: Amanda Rebar

Institution and Country: Central Queensland University, Australia

Please state any competing interests: None declared

Point 1:

I think the state of the literature is such that a cross-sectional test of the association of physical activity (even objectively monitored) and depressive symptoms no longer advances our knowledge. There is quite a bit already known about the association of physical activity and depressive symptoms (see Cooney et al. 2014 Cochrane review; and Rebar et al., 2016 meta-meta-analysis in Health Psychology Review), which includes several studies with similar populations. However, there is exciting potential for this study to provide more rich evidence of prospective, intervention or mediation evidence relevant for the link between activity and depression in people at risk for CVD. For example,

previous studies have shown that the association between depressive symptoms and adverse CVD events was explained by physical inactivity (Whooley et al., 2008, JAMA). By using more of the RCT data points, this study could provide for important evidence

Reply:

We are fully aware that there are a number of cross-sectional studies on the association of PA and behavioural or psychological factors. However, these studies have been conducted on small non-European samples or in specific disease cohorts. To the best of our knowledge, no study has examined a large European sample of healthy older adults at high risk of CVD. The Cochrane review and the meta-meta-analysis by Rebar et al., 2015 synthesised findings on the effectiveness of exercise as treatment for depression but not the relationship between levels of activity and levels of depression. From the perspective of clinicians, it is important to understand that while exercise interventions might not be more effective than usual treatment for depression (Cooney et al., 2013), depressive symptoms are in fact associated with lower step count. Especially when working with people at high risk of CVD, we need to raise awareness that depressive symptoms are not only associated with worse CVD outcomes, but also adverse health behaviour. We have provided exceptionally high quality physical activity data in a large sample and as a result improved the validity of previous observations thus making a valuable contribution to the existing evidence base. We also added the meta-meta-analysis to our discussion, as it was a good addition (lines 371 ff.).

Point 2:

The conclusion of the abstract includes a statement that 'this may potentially modify the effectiveness of lifestyle interventions to reduce the risk of CVD.' This seems a bit of a stretch based on the data presented in this study. Please consider a conclusion more closely tied to the study findings.

Reply:

This is a reasonable point. This sentence was changed to 'People at high risk of CVD with depressive symptoms have lower levels of physical activity.' (lines 51 to 52).

Point 3:

Given that the measure of depressive symptoms is continuous, why not test it as a continuous measure instead of separated into categories based on scores?

Reply:

This was indeed an issue we thought about and we are aware that we may lose power by categorizing the score. However, had we entered the PHQ-9 score as a continuous measure, we could have detected changes in step count per unit in-/decrease. Considering that a) not every point increase in the PHQ-9 sum score translates into a clinically relevant change and b) our sample did not cover the total PHQ-9 sum score range from 0 to 27 and is thus more prone to outliers in the higher range, we decided to simplify the model by focussing on reporting clinically distinctive subgroups that are better represented by our data. Alternatively, a non-linear relationship between the continuous score and the ln-transformed number of steps would have needed to be modelled, which would have involved a lot of complex model selection in addition to the already very complex model. We tried to make it as easy and straightforward as possible by avoiding more extensive modelling.

Point 4:

The point about people with more depressive symptoms tended to wear the monitors for less days than people without depressive symptoms is an important one and the authors should be commended for considering the potential bias this may cause.

Reply:

Thank you.

Point 5:

Please clarify what is meant by 'merging' the step counts for weekend and weekdays.

Reply:

'merging' meant calculating the arithmetic means, we changed this in the respective section (lines 194 to 196).

Point 6:

For the t-tests, adjustments to the p-values are needed for the multiple test comparisons.

Reply:

Table 1 shows tests to identify important confounders to avoid including too many variables into our model. Using a lower alpha value would increase the risk to miss confounders for our final model (by increasing the type II error and falsely keeping the Null-Hypothesis of "There is no confounding effect" (Rothman, 1990) and we, therefore, did not use correction for multiple testing (lines 198-202).

However, in our final model results multiple testing can be an issue and we therefore lower the alpha level to 0.01. Predictors with a p-value between 0.01 and 0.05 will be treated as a trend, which should be interpreted with care (Lang & Secic, 2006). We mention this in lines (224-225).

Point 7:

The value of the categories for the covariate variables need to be presented.

Reply:

We have already summarised the values of the categories of the covariates in Table 1.

Point 8:

Was there nesting at the level of clinic that should perhaps be accounted for? Check the intraclass correlations for that.

Reply:

We assessed the variance of clinic by including clinic as a random effect in our model. The estimated variance was virtually 0 and we therefore excluded clinic from the model. We added this information to the method section (lines 219-220).

Point 9:

The discussion often talks about change (e.g., 'decrease' 'worsening'), which does not reflect the cross-sectional association found.

Reply:

You are right, we changed references that infer causality ('decrease' 'worsening') in the discussion (lines 290 ff.).

Reviewer: 3

Reviewer Name: Dr Caroline Anne Mitchell

Institution and Country: ACADEMIC UNIT OF PRIMARY MEDICAL CARE, UNIVERSITY OF SHEFFIELD

Please state any competing interests: NONE DECLARED

Please leave your comments for the authors below

Overall this a well written, novel and interesting study investigating the interplay between physical activity and depression within a group of people at higher risk of cardiovascular events.

The limitations section should be strengthened with more commentary on the biases of the study.

Point 1:

spelling error in abstract: practises should be 'practices' plural of practice noun not verb

Reply:

Changed, thank you.

Point 2:

The abstract is clear and the method used to explore associations robust

Reply: Thank you.

Point 3:

ERROR IN TABLE OR MAIN TEXT RE GENEDED BALANCE OF THE SAMPLE

Population characteristics: authors to check the statement the 'vast majority were male' made within this paragraph as Table 1 (total population characteristics) states that 85.5% + were female and 14.5% were men. I think the table must be incorrect as in the discussion, the explanation for the gender bias of the sample is that women are less likely to have a raised q-risk. If table 1 is incorrect then the reader needs to be assured that the rest of the tables are correct in relation to the statistical analyses by gender as a confounder.

Reply:

Thank you for noticing this mistake. Gender was swapped in Table 1. The results for gender as confounders in the subsequent analyses are correct.

Point 4:

The limitations section I think should be expanded and more explicitly acknowledge the biases within a population derived from a sample recruited to a RCT: These limitations include:

a. the generalizability of the sample. The observations are from the baseline data of participants in a RCT. There is therefore risk of participation bias as the sample participating in the study will differ to an unmeasured extent from the rest of the primary care population within the capital city of the UK. This population may well already be more likely to be interested in exercise having consented to participate in an activity monitoring behaviour change study.

Reply:

We have included the participation bias in the limitations (lines 309-312).

Point 5:

The rationale to merge all categories of depression with a PHQ9 > 10 into a single category was necessary as a relatively small number of people self reported potentially clinically significant depression with PHQ9 score > 10. Thus the total within the potentially clinically significant depression category was 4% of the total population.

Reply:

This is a correct interpretation.

Point 6:

c. Population characteristics- Table 1 states that 8.5 5%+ were women, almost 89.5% were white which does not reflect the general population of the UK, and especially not the ethnic diversity of London.

So the following statement, I think should be changed to acknowledge this bias

The statement: is representative of the ethnic, socioeconomic, urban and semi-rural diversity typical of large cities in the developed nations.

As 2011 census data states that :

London had a population of 8,173,941. Of this number, 44.9% were White British. 37% of the population were born outside the UK, including 24.5% born outside of Europe.

Reply:

We have now removed the statement.

Reviewer: 4

Reviewer Name: Patricia Wong

Institution and Country: University of Pittsburgh, USA

Please state any competing interests: None declared

Please leave your comments for the authors below

The authors present an interesting and relevant question: does depression predict low physical activity among those at risk for cardiovascular disease? Because increasing physical activity can help reduce this risk, understanding what may hinder improved activity levels can have important implications for treatment (i.e., targeting depression if it is a risk factor). There are several comments and questions this reviewer has concerning the methods and interpretation of results:

Point 1:

A strength of the study is the large sample size and activity measured via the Actigraph. In addition, the authors take into account several important control variables. One variable not considered is sleep-- do the authors have access to information regarding participants sleep characteristics (duration, quality, timing, etc) either measured via the same Actigraph data or through other measures? Because sleep disturbances can be related to depression, physical activity, and cardiovascular disease risk, it'd be important to also consider this additional control variable.

Reply:

Unfortunately, sleep was not measured, but this is a good addition to future studies.

Point 2:

The "Implications for clinicians" statement is a bit unclear. The authors conclude that depression may be an important factor to consider when treating those with cardiovascular risk because it may hinder interventions focused on improving physical activity. However, they cite evidence that shows targeting physical activity does not improve depression symptoms. This rationale, at least as written, does not support the conclusion: ineffectiveness of physical activity interventions to treat depression does not indicate that depression hinders the treatment of physical activity. As the authors noted, their data is cross-sectional so any directionality between depression, physical activity, and effectiveness of treatment for cardiovascular risk cannot be assumed. Is there other previous literature the authors can draw from to support their conclusion?

Reply:

You are right, this was not well stated. We cannot draw the inverse conclusion that if PA interventions are ineffective in treating depression, depression may be barrier to increasing PA. We changed the paragraph to only state what we can draw from the existing evidence (line 371 to 380).

References

Cooney, G. M., Dwan, K., Greig, C. A., Lawlor, D. A., Rimer, J., Waugh, F. R., ... Mead, G. E. (2013). Exercise for depression. <https://doi.org/10.1002/14651858.cd004366.pub6>

French, M. T., Popovici, I., & Maclean, J. C. (2009). Do Alcohol Consumers Exercise More? Findings from a National Survey. *American Journal of Health Promotion*, 24(1), 2–10. <https://doi.org/10.4278/ajhp.0801104>

Lang, T. A., & Secic, M. (2006). *How to report statistics in medicine: annotated guidelines for authors, editors, and reviewers* (2nd Edition). ACP Press.

Paykel, E. S., Brugha, T., & Fryers, T. (2005). Size and burden of depressive disorders in Europe. *European Neuropsychopharmacology*, 15(4), 411–423. <https://doi.org/10.1016/j.euroneuro.2005.04.008>

Rothman, K. J. (1990). No adjustments are needed for multiple comparisons. *Epidemiology* (Cambridge, Mass.), 1(1), 43–46.

Walters, P., Barley, E. A., Mann, A., Phillips, R., & Tylee, A. (2014). Depression in Primary Care Patients with Coronary Heart Disease: Baseline Findings from the UPBEAT UK Study. PLoS ONE, 9(6), e98342. <https://doi.org/10.1371/journal.pone.0098342>

Westertep, K. R., Meijer, E. P., Goris, A. H. C., & Kester, A. D. M. (2004). Alcohol energy intake and habitual physical activity in older adults. British Journal of Nutrition, 91(01), 149. <https://doi.org/10.1079/BJN20031013>

VERSION 2 – REVIEW

REVIEWER	Caroline Vandeleur Lausanne University Hospital, Department of Psychiatry
REVIEW RETURNED	05-Feb-2018

GENERAL COMMENTS	The authors have sufficiently revised their manuscript and I recommend publication
--

REVIEWER	Patricia Wong University of Pittsburgh, USA
REVIEW RETURNED	08-Feb-2018

GENERAL COMMENTS	The authors have sufficiently addressed all of this reviewer's previous comments/concerns.
--

REVIEWER	Amanda Rebar Central Queensland University, Australia
REVIEW RETURNED	08-Feb-2018

GENERAL COMMENTS	Thanks for the comments and revisions. I have no further concerns with the manuscript.
--

REVIEWER	CAROLINE MITCHELL Academic Unit of Primary Medical Care , University of Sheffield
REVIEW RETURNED	23-Feb-2018

GENERAL COMMENTS	Reviewer Feedback has been addressed within this submitted version
--